# Proton and Carbon Ion Irradiation Changes the Process of Endochondral Ossification in an Ex Vivo Femur Organotypic Culture Model

**DOI:** 10.3390/cells12182301

**Published:** 2023-09-18

**Authors:** Vanessa Etschmaier, Dietmar Glänzer, Nicole Eck, Ute Schäfer, Andreas Leithner, Dietmar Georg, Birgit Lohberger

**Affiliations:** 1Department of Orthopaedics and Trauma, Medical University Graz, 8036 Graz, Austria; vanessa.mair@medunigraz.at (V.E.); dietmar.glaenzer@medunigraz.at (D.G.); nicole.eck@medunigraz.at (N.E.); andreas.leithner@medunigraz.at (A.L.); 2Department of Neurosurgery, Research Unit for Experimental Neurotraumatology, Medical University of Graz, 8036 Graz, Austria; ute.schaefer@medunigraz.at; 3Department of Radiation Oncology, Medical University of Vienna, 1090 Vienna, Austria; dietmar.georg@meduniwien.ac.at; 4MedAustron Ion Therapy Center, 2700 Wiener Neustadt, Austria

**Keywords:** organotypic bone slides, particle therapy, proton irradiation, carbon ion irradiation, bone biology

## Abstract

Particle therapy (PT) that utilizes protons and carbon ions offers a promising way to reduce the side effects of radiation oncology, especially in pediatric patients. To investigate the influence of PT on growing bone, we exposed an organotypic rat ex vivo femur culture model to PT. After irradiation, histological staining, immunohistochemical staining, and gene expression analysis were conducted following 1 or 14 days of in vitro culture (DIV). Our data indicated a significant loss of proliferating chondrocytes at 1 DIV, which was followed by regeneration attempts through chondrocytic cluster formation at 14 DIV. Accelerated levels of mineralization were observed, which correlated with increased proteoglycan production and secretion into the pericellular matrix. Col2α1 expression, which increased during the cultivation period, was significantly inhibited by PT. Additionally, the decrease in ColX expression over time was more pronounced compared to the non-IR control. The chondrogenic markers BMP2, RUNX2, OPG, and the osteogenic marker ALPL, showed a significant reduction in the increase in expression after 14 DIV due to PT treatment. It was noted that carbon ions had a stronger influence than protons. Our bone model demonstrated the occurrence of pathological and regenerative processes induced by PT, thus building on the current understanding of the biological mechanisms of bone.

## 1. Introduction

Radiation therapy (RT), together with chemotherapy and surgery, is one of the most commonly used therapeutic tools to treat a wide range of malignancies [1]. However, radiation also has adverse effects on bone, including disturbances in bone growth, osteoradionecrosis, pathologic fracture, and radiation-induced neoplasms [2]. Furthermore, a clear dose–response relation exists, whereby higher doses lead to higher levels of toxicity, which are more prevalent in growing as opposed to mature bone. Long-term follow-up examinations revealed that pediatric oncology patients treated with RT suffer from major growth disturbances. This includes the asymmetry of chest and ribs, scoliosis, kyphosis, pain at the radiation site, leg-length inequality, and in some cases the development of secondary tumors [3,4].

Postnatal bone development is accomplished through endochondral ossification when growth plate chondrocytes undergo a highly organized and structured differentiation and maturation process. Here, the chondrocytes mature from a resting to a proliferation state and finally become hypertrophic, which ends in cell death and matrix calcification [5]. This highly complex process is subject to tight regulation by transcription factors, cytokines and extracellular matrix proteins. In the differentiation of mesenchymal cells into chondrocytes or osteoblasts, transcription factors such as runt-related transcription factor 2 (Runx2), in particular, play a crucial role in ensuring the binding and differentiation of mineralizing cells. In addition, the expression of Runx2 in pre-hypertrophic chondrocytes is necessary to promote terminal hypertrophy [6].

In the cartilaginous extracellular matrix (ECM) of the growth plate, type 2 collagen alpha 1 chain (Col2α1) forms a network with proteoglycan. This matrix of cartilage is able to bind growth factors, hyaluronic acid, and other molecules in response to mechanical stress or in order to maintain physiological homeostasis [7]. To allow bones to grow longitudinally, chondrocytes use their ability to increase their cell volume 10-fold before going into apoptosis [8]. Once they become hypertrophic, chondrocytes secrete collagen type 10 (ColX) into the ECM, which is a key marker used to detect chondrocyte hypertrophy [9].

After exposing growing bone to radiation, radiation-induced acute cell death, proliferative arrest, and accelerated differentiation and maturation of chondrocytes occurs, which are then subsequently lost for the ongoing bone elongation process [10,11,12]. Particle therapy (PT) with protons or carbon ions (C-ions) for bone tumors, can improve this local control mechanism and increase patient survival rates [13], therefore representing an essential therapeutical option. The main argument for proton irradiation (IR) compared to conventional photon therapy (X-ray) is its dosimetric ballistics, i.e., the dose rises sharply at a well defined depth (Bragg peak) and falls rapidly beyond this maximum. In this way, a highly conformal high-dose region can be tailored to cover the target volume with high precision, while reducing the burden on surrounding healthy tissue [14]. C-ions have a higher linear energy transfer (LET), contributing to higher relative biological effectiveness (RBE). As a result, it is generally considered the most efficient treatment for radioresistant bone tumors [15].

Although the side effects of clinical PT use are well known, there is a lack of comprehensive preclinical research providing a better understand of the basic cellular radiation response mechanisms. Commonly, research relies on methods like conventional in vitro two-dimensional (2D) cell culture or animal studies to explore the research question. Two-dimensional cell cultures are fast, cost-effective and have high-throughput testing, thus representing a fundamental tool in basic research. However, the gathered data are often not representative of a complex biological system, therefore requiring further in vivo animal studies. To overcome the challenge of poor translation and to adhere to the 3Rs (“Reduce, Refine and Replace”) of animal studies, 3D organotypic cultures (OTCs) have been developed. With the development of such ex vivo organotypic bone cultures, which mimicked in vivo biological conditions, a new methodology for bone regeneration study was achieved [16,17]. In this study, we aimed to establish and characterize the effects of proton and C-ion IR on an ex vivo proximal femur OTCs from Sprague Dawley rats using histology, immunohistochemistry, and gene expression analysis.

## 2. Materials and Methods

### 2.1. Experimental Design

For the preparation of 300 µm ex vivo bone slices, femurs were taken from 4-day-old Sprague Dawley rats. Slices underwent 8 Gy proton or C-ions IR. After 1 and 14 days of in vitro (DIV) cultivation, bones slices were harvested for subsequent analysis. The workflow is presented in Figure 1a. To ensure the robustness and reliability of the obtained results, experiments were conducted the experiment 4 times for gene expression analysis (16 rat pups in total) and 3 times for histological and immunohistochemistry staining (12 rat pups in total) with 3 sections per OTC at different dorso-ventral depths.

### 2.2. Preparation and Culturing of Ex Vivo Organotypic Bone Slice Culture (OTCs)

Rat femurs were quickly explanted, washed in ice-cold Dulbecco’s phosphate-buffered saline (DBPS) (Thermo Fisher Scientific, Waltham, MA, USA) and processed into an agar block. Next, 300 µm coronal bone slices were produced with an VT1000 Leica vibratome (Leica Biosystems, Wetzlar, Germany) according to the manufacturer’s instructions, under buffered conditions using ice-cold 1× DPBS (Thermo Fisher Scientific) and a stainless-steel razor blade (Gillette Procter & Gamble, Cincinnati, OH, USA) at an amplitude of 1.5 mm and sectioning speed of 0.06 mm/s using a consistent blade angle. Afterwards, the sample was gently washed in minimum essential medium eagle (MEM) (Sigma-Aldrich, St. Louis, MO, USA) supplemented with penicillin (100 U/mL) and streptomycin (100 µg/mL) (Invitrogen, Thermo Fisher Scientific) for 30 s and transferred into 6-well plates containing a Millicell cell culture 30 mm insert with pore size 0.4 µm (MERCK KGaA, Darmstadt, Germany) for subsequent cultivation. Each well was supplemented with 1 mL of osteogenic medium containing MEM with 5% horse serum (Invitrogen, Thermo Fisher Scientific), 1% penicillin/streptomycin, 1× Insulin–Transferrin–Selenium (Gibco, Invitrogen, Waltham, MA, USA), 100 μM ascorbic acid, 10 mM β-glycerophosphate (both Sigma-Aldrich, St. Louis, MI, USA), 10 nM dexamethasone (Stemcell Technologies, Vancouver, BC, Canada), and 0.01 µg/mL TGFβ-3 (Thermo Fisher Scientific). On a daily basis, 500 µL of medium was replaced with fresh medium. Six-well plates were incubated in a humidified atmosphere at 37 °C with 5% CO_2_. Slices were analyzed on 1 DIV and 14 DIV.

Male and female adult Sprague Dawley rats were obtained from Charles Rivers, housed and bred under standard conditions with 12 h:12 h light–dark cycle and ad libitum access to food and water in the animal facility of Biomedical Research (BMF) at the Medical University Graz in accordance with §25 of Animal Testing Act 2012 (TVG 2012).

### 2.3. Physical Parameters of Irradiation (IR)

All IR experiments were conducted at MedAustron, the Austrian center for ion therapy and research that relies on synchrotron technology [18]. Within the experimental research room, a horizontal beam line equipped with active spot scanning technique and active energy variation for both proton and C-ions was used. The precise and standardized positioning of samples embedded in measurement phantoms was made possible by utilizing a high precision robot couch and a laser positioning system. For the proton IR, a treatment plan was created using a 4 cm spread-out Bragg peak (SOBP) and a field size of 17 × 9 cm^2^, with the assistance of the RayStation v7.99 treatment planning system (TPS) from RaySearch Laboratories (Stockholm, Sweden). The dose calculation was performed using a Monte Carlo v4.3 dose engine [19]. The maximum proton energy employed was 124.7 MeV, resulting in a range of 101.2 mm at a 80% dose level in water. Likewise, the C-ion IR involved a treatment plan with a 4 cm SOBP, using the same field sizes, TPS, and dose engine. The maximum C-ion energy utilized was 238.6 MeV/u, with a range of 101.1 mm at the 80% dose level in water. Ripple filters were applied to ensure a flat SOBP. The energy layers for both types of particles were spaced either 1 mm or 2 mm apart. The radiation delivery technique employed was pencil beam scanning, and no range shifters were used to modify the beam. The linear energy transfer (LET) values for both proton and C-ion IR were directly determined from the TPS using Monte Carlo calculations. Subsequently, bone slices of the right leg were irradiated with either 8 Gy of proton or 8 Gy of C-ions, while the bone slices of the corresponding left leg remained unirradiated, serving as control samples. The selection of the 8 Gy dosage was made to effectively demonstrate its impact on postnatal bone elongation within the proximal femur epiphysis.

### 2.4. Tissue Processing for Histological Evaluation and Analyzes

The bone slices were fixed in 4% paraformaldehyde (PFA; MERCK KGaA) using 2 mL per slice, and left overnight at 4 °C. Subsequently, the fixed slices were embedded in paraffin without decalcification using a VIP 5E-F2 Tissue Processor (Sakura, Tokyo, Japan). Serial sections of 5 µm thickness were obtained using the Microtome HM 360 rotation microtome (Hyland Scientific GmbH, Berlin, Germany). For histological staining and immunohistochemistry, the sections underwent overnight incubation at 60 °C, followed by de-paraffinization and rehydration processes. Whole object slides were scanned with an Aperio ScanScope (Leica Biosystems, Wetzlar, Germany), and the images were acquired using Image Scope software (v12.4.0.50.43; Leica Biosystems, Deer Park, IL, USA). Morphological changes were assessed through H&E staining, while tissue architecture analysis was carried out using Movat’s Pentachrome staining. The staining process involved the following steps: de-paraffinized and rehydrated bone sections were stained with a 0.1% Alcian blue solution (CarlRoth, Karlsruhe, Germany) and then stabilized in 10% alkaline ethanol (CarlRoth). Subsequently, staining was performed with Weigert’s iron hematoxylin (A + B 1:1, CarlRoth), brilliant crocein (Sigma-Aldrich, St. Louis, MI, USA), and acid fuchsin (CarlRoth) solutions. The sections were differentiated with phosphotungstic acid (CarlRoth) and finally stained with Gatinais saffron staining (Chroma-Waldeck, Münster, Germany) to complete the protocol. The various structures in the stained sections were represented by different colors: cell nuclei appeared black, cytoplasm appeared reddish, elastic fibers appeared red, collagen connective tissue appeared light yellow, mineralized tissue/bone appeared bright yellow, and mineralized cartilage appeared green, as previously demonstrated by Sun et al. [20].

### 2.5. Immunohistochemistry (IHC) Staining

To gain further knowledge on the distribution of the extracellular matrix proteins, as well as chondrogenesis and osteogenesis markers, IHC was performed with the kit Ultra Vision LP/HRP rabbit/mouse (Epredia, Kalamazoo, MI, USA). Antigen retrieval was performed with either 1 mM EDTA (pH 8.0) or 10 mM sodium citrate buffer (pH 6.0) with 0.05% Tween-20. After 0.3% hydrogen peroxidase (H_2_O_2_) and protein block, primary antibodies Col2α1 (1:300, Invitrogen), ColX (1:100, Invitrogen), BMP-2 (1:100, Invitrogen), Runx2 (1:100, Abcam, Cambridge, UK), OPG (1:250, Abcam), and SPP1 (1:250, Novus biologicals, Littleton, CO, USA) were incubated over night at 4 °C. Primary antibodies from mice required a bridging step, before all sections were incubated with HRP-enzyme. Positive staining was developed with the AEC Substrate kit (Abcam) followed with a hematoxylin counter stain.

### 2.6. RNA Preparation and Reverse Transcription Polymerase Chain Reaction (RT-PCR)

After IR, the bone slices were cultured for 1 and 14 DIV, respectively. After these time frames, the bone slices were shock frozen in liquid nitrogen and tissue pulverization was achieved with mortar and pestle on dry ice with liquid nitrogen. Total RNA was isolated using the RNeasy Mini Kit and DNase-I treatment according to the manufacturer’s manual (Qiagen, Hilden, Germany). Two µg RNA were reverse transcribed with the iScript-cDNA Synthesis Kit (BioRad Laboratories Inc., Hercules, CA, USA) using a blend of oligo(dT) and hexamer random primers. Amplification was performed with the SsoAdvanced Universal SYBR Green Supermix (Bio-Rad Laboratories Inc.) using technical triplicates and measured by the CFX96 Touch (BioRad Laboratories Inc.). Results were analyzed using the CFX manager software (version 3.1) for CFX Real-Time PCR Instruments (Bio-Rad Laboratories Inc.). Next, quantification cycle values (Ct) were exported for statistical analysis. Results with Ct values greater than 32 were excluded from the analysis. The relative quantification of expression levels was obtained by the ∆∆Ct method based on the geometric mean of the internal controls hypoxanthine phosphoribosyltransferase 1 (HPRT) and glyceraldehyde-3-phosphate dehydrogenase (GAPDH), respectively. The expression level (Ct) of the target gene was normalized to the reference genes (ΔCt); the ΔCt of the test sample was normalized to the ΔCt of the control (ΔΔCt). Finally, the expression ratio was calculated with the 2^−ΔΔCt^ method. The sequences of all primers used are listed in the Appendix A.

### 2.7. Data Presentation and Statistical Analyses

The results are presented as the mean  ±  SEM of independent variables, each obtained from quantification in at least four experimental replicates. Statistical analyses were performed using IBM SPSS Statistic 26 (version 6, New York, NY, USA) and GraphPad InStat software (GraphPad Prism 9.3.1 (471), Boston, MA, USA). The normality of the data was assessed using the Kolmogorov–Smirnov test. As the data distribution in all samples significantly deviated from normal distribution, non-parametric tests were used to test the statistical significance of the observed differences. Single comparisons were analyzed using the Mann–Whitney U-Test, while multiple comparisons were tested with Kruskal–Wallis H test followed by pairwise analysis with Bonferroni correction. A difference with *p* ≤ 0.05 was considered statistically significant in all statistical assessments.

## 3. Results

### 3.1. Morphological Changes on Endochondral Ossification Epiphysis after Particle Irradiation

The postnatal distal epiphysis was divided into three distinctive zones—transition zone (TZ), middle zone (MZ) and deep zone (DZ)—depending on the maturation and differentiation stage of chondrocyte and their cell organization (Figure 1b). To examine the effect of radiation on endochondral ossification, ex vivo bone cultures with their respective controls after 1 and 14 DIV cultivation were histologically analyzed with H&E and Safranin-O/Fast Green staining (Appendix A). To ensure a direct comparison between protons and C-ions, a dose of 8 Gy was selected for both particle types.

H&E staining was performed to assess the radiation-induced morphological changes in ex vivo bone cultures (Figure 2). One day after IR, the presence of randomly distributed pre-hypertrophic chondrocytes within the TZ of bone cultures exposed to 8 Gy proton IR were compared to the respective non-IR controls. Within the MZ for both, proton and C-ions IR, a profound loss of proliferating chondrocytes was witnessed and replaced by pre-hypertrophic and hypertrophic chondrocytes. Moreover, chondrocytes in the DZ were not affected after 1 DIV cultivation. After 14 DIV, the effects of IR could be observed and were most pronounced in chondrocytes within the MZ. Bone cultures exposed to 8 Gy proton IR became pre-hypertrophic and/or hypertrophic and arranged into columns. This is usually a characteristic for the DZ and necessary for the bone elongation process. After C-ions IR, chondrocytes generally increased in size, formed isogenic groups, but also became pre-hypertrophic within these groups.

### 3.2. Particle Irradiation Affected Endochondral Ossification

Movat’s pentachrome staining was performed to assess the process of endochondral ossification after IR and subsequent in vitro cultivation (Figure 3). No profound differences were observed after the 1 DIV cultivation of the radiated bone compared to the non-IR controls. After 14 DIV, bone cultures exposed to proton IR exhibited tissue mineralization (bright yellow) within the TZ compared to the respective control. On the contrary, cartilage is the predominant tissue within C-ions IR bone cultures (green). In the MZ, cartilage (green) is predominantly presented after proton IR, whereas C-ions IR cultures had a similar distribution as seen in the control. Both IR types revealed an increased mineralization of the cells (bright yellow) next to the zone of Ranvier in the MZ. Proton IR provoked a more intense cartilage staining within the DZ, while C-ions IR triggered a reduction (Figure 3 D). Notably, mineralized cells within the MZ of 14 DIV were positively stained for proteoglycans. Here, both IR types exhibited more cells intensely stained for proteoglycans than the control (Appendix A).

### 3.3. Effect of Particle Irradiation on Extracellular Matrix (ECM) Composition

To gain further understanding on the variation of tissue composition observed with the respective zones, Movat’s pentachrome staining was undertaken. Next, the expressions of Col2α1 and ColX by IHC as well as their gene expressions were analyzed (Figure 4). Respective IHC controls are presented in the Appendix A. An overview of the distal epiphysis revealed progressive endochondral ossification within the in vitro cultivation frame, indicated by mineralization of cartilage forming a ring within the epiphysis excluding the later secondary ossification center (green). Pronounced mineralized bone was observed towards the later articular cartilage (bright yellow). Moreover, radiated bone slices also exhibited areas of mineralized bone within the TZ at 14 DIV. At this time point, collagen connective tissue was observed at the area, becoming the secondary ossification center (light yellow-greenish staining) (Figure 4a).

With regard to Col2α1, only moderate staining was observed within the columnar zone of the DZ and towards the later secondary ossification center at 1 DIV (Figure 4b). This increased profoundly during the cultivation period. At 14 DIV, the Col2α1 spread from the center of the epiphysis towards the MZ and DZ. Here, the control slices exhibited the most positive stained area followed by C-ions and proton IR. This observation was also strengthened by the gene expression analysis. For all conditions, a highly significant upregulation was observed from 1 DIV to 14 DIV (non-IR: 5.82 ± 4.3 ***; proton: 5.33 ± 3.0 ***; C-ions: 2.92 ± 1.3 ***). At 14 DIV, C-ions IR had a significant impact of the Col2α1 expression compared to the non-IR controls (**) (Figure 4d).

Chondrocyte hypertrophy is essential for bone elongation and is marked by the ECM protein ColX. At 1 DIV, the control and proton IR slices exhibited moderately positive staining of the inter-territorial matrix of hypertrophic chondrocytes within the DZ (Figure 4c). The height of hypertrophic area is decreased after proton IR. Within C-ions, radiated bone slices ColX is confined within the chondrocytes and is not yet secreted to the ECM at this early time point. After 14 DIV, a profound increase in ColX staining intensity was observed within all conditions. However, PT led to a lower extent of the hypertrophic area. Notably, at the site of calcification, the proton radiated bone slices demonstrated an intensely positive ColX staining. Moreover, the gene expression data demonstrated a decrease in ColX expression due to radiation at 1 DIV with a highly significant decrease in gene expression from 1 DIV to 14 DIV for all conditions, respectively (non-IR: 0.14 ± 0.1 ***; proton: 0.04 ± 0.1 ***; C-ions: 0.01 ± 0.01 ***). At 14 DIV, a significant difference was observed between the control and proton IR (*) and C-ions (***) (Figure 4e).

### 3.4. Cartilage Regeneration and Progression of Chondrocyte Maturation under the Influence of Particle Irradiation

Next, cartilage regeneration and the progression of chondrocyte maturation (chondrogenesis) were investigated. Bone morphogenetic protein 2 (BMP-2), Runx-2, and osteoprotegerin (OPG) were analyzed by IHC as well as gene expression by RT-PCR. Overview images of the respective zones in a higher magnification are presented in the Appendix A (BMP-2), Appendix A (Runx2), and Appendix A (OPG).

BMP-2 is known to induce chondrogenesis as well as promote endochondral ossification, and was used to assess the regeneration potential. Only a light positive staining was observed within the TZ, MZ, and DZ for all conditions at 1 DIV (Figure 5a). This was especially the case after C-ions IR, as a loss of positive staining was observed within these regions. After 14 DIV, chondrocytes were more intensely stained for BMP-2 under all conditions. Within the TZ, a reduction in positive staining was observed due to C-ions IR. A decrease in positive chondrocyte staining was observed within the MZ and DZ for both PT conditions, respectively. Interestingly, the BMP-2 gene expression revealed a significant upregulation at 1 DIV after proton IR (4.76 ± 4.9 *) and a significant downregulation after C-ions IR (0.74 ± 0.3 **) (Figure 5d). For all conditions, a significant upregulation after 14 DIV was observed (non-IR: 9.98 ± 4.2 ***; proton: 6.33 ± 3.8 ***; C-ions: 2.75 ± 2.3 *). Although gene expression is upregulated during the duration of cultivation, proton IR (*) and C-ions IR (***) after 14 DIV caused a significant downregulation of BMP2 (Figure 5d).

We then proceeded to analyze Runx2, a transcription factor that is essential in promoting the terminal hypertrophy of chondrocytes. At 1 DIV, a decreased positive staining of the chondrocytes within the TZ and MZ was observed due to PT (Figure 5b). While proton IR had no effect on the positive staining of chondrocytes within the DZ, for C-ions IR, an increase in Runx2 positive chondrocyte staining was observed. At 14 DIV, a decreased staining due to PT was observed within the TZ and MZ. Here, positive stained chondrocytes were randomly distributed and lost their organized structure, as seen in the non-IR control. In the DZ, no profound difference of Runx2 staining after PT was observed. PT had a significant impact on the Runx2 gene expression at 1 DIV (proton: 0.65 ± 0.3 ***; C-ions: 0.64 ± 0.3 ***). After the 14 DIV gene expression was upregulated for all conditions (non-IR: 4.69 ± 5.9 **; proton: 1.76 ± 1.6; C-ions: 1.04 ± 0.5). Runx2 gene expression was significantly downregulated due to PT at 14 DIV (proton: *; C-ions: **) (Figure 5e).

At 1 DIV, chondrocytes positively stained for OPG were restricted to the TZ and MZ of the control, while they are found throughout the epiphysis (TZ, MZ, and DZ) due to PT (Figure 5c). After 14 DIV, a decrease in staining was observed in the TZ, but an increase in the DZ and bone marrow. Detailed images of all zones are shown in Appendix A. OPG gene expression analysis revealed no significant differences at 1 DIV due to PT. At 14 DIV, a significant upregulation for all conditions was observed (non-IR: 8.21 ± 6.0 ***; proton: 9.84 ± 3.4 ***; C-ions: 5.41 ± 1.65 ***). A significant reduction in OPG gene expression was seen due to C-ions IR (*) (Figure 5f).

Osteogenesis was analyzed with the IHC of mature osteoblast marker osteopontin (SPP1) as well as gene expression analysis of SPP1 and alkaline phosphatase (ALPL). At 1 DIV, a light positive staining of SPP1 within the periosteum after PT was observed (Figure 6a). At 14 DIV the periosteum was intensely positive stained within all conditions. The gene expression levels further revealed a highly significant downregulation of SPP1 due to the proton IR at 1 DIV (0.61 ± 0.3 ***). During the 14 DIV, all conditions were observed to be highly upregulated (non-IR: 11.66 ± 8.8 ***; proton: 14.81 ± 6.2 ***; C-ions: 10.5 ± 7.2 ***); however no significant differences due to PT were detected (Figure 6b). At 1 DIV, IR induced significant downregulation of ALPL gene expression (proton: 0.57 ± 0.3 ***; C-ions: 0.69 ± 0.2 ***) (Figure 6c). According to the progression of osteogenesis, ALPL expression significantly increased after 14 DIV, whereas the PT inhibited the expression (non-IR: 2.62 ± 1.2 ***; proton: 1.28 ± 0.7; C-ions: 1.40 ± 1.1). Compared to the non-IR control, a significant reduction by proton (***) and C-ions IR (**) was observed.

## 4. Discussion

Radiation therapy (RT) is one of the gold standard treatment methods for malignant tumors [1]. Common side effects include skin redness, hair loss, radiation burns, or acute radiation syndrome [21]. Due to its special physical and biological characteristics, particle therapy (PT) enables a more targeted application of the high-energy radiation to the tumor, while protecting the surrounding tissue as much as possible [19]. Within the so-called “Bragg peak”, the accelerated particles discharge most of their energy capacity directly into the target tissue [22]. While protons deliver all of their energy within the Bragg peak, C-ions undergo fragmentation and only deliver a fraction of dose to the distal tissue [23]. Furthermore, C-ions possess an elevated RBE, which is related to the higher LET, i.e., the higher ionization density and track structure of secondary particles resulting from interactions after IR [24]. Although PT is both more complex and significantly expensive, the advantages offered to the adjacent healthy tissue and/or tumor tissue contribute to PT becoming the preferred irradiation methods for numerous radioresistant and pediatric tumors [21]. In clinical practice, the particles are applied in the following dosages: 70 Gy in 35 fractions at 2 Gy RBE/fraction (proton IR) and 70.4 Gy in 16 fractions at 4.4 Gy RBE/fraction (C-ions). The irradiation of cells or bones for research purposes is typically performed with doses of 2 Gy, 4 Gy, or 8 Gy for both proton and C-ion irradiation. Due to the limited availability of irradiation time for research, which is usually limited to weekends when the particle accelerator is not required for patients, the 8 Gy dosage was chosen in this particular case to clearly demonstrate the effects being investigated.

A fundamental concern when treating pediatric patients is the ongoing process of endochondral ossification, as well as post-irradiation complications, which can occur for both RT or PT [25]. While the adverse lifelong effects of RT on the growth plate (GP) have been well described [10,11,26], those of PT on GP have not been investigated to date. This is due to the limited availability of adequate research infrastructure. It is well described that RT has numerous adverse effects on growing bone and skeletal development. Post-IR complications include growth arrest, growth retardation, fracture, or osteoradionecrosis and are seen with doses as low ≥15 Gy; however, the most clinically relevant measurable effects are seen at doses beyond 30 Gy [26]. We observed a severe loss of proliferating chondrocytes within the TZ after one single dose of PT (8 Gy proton or C-ions IR) at 1 DIV. After 14 DIV, proliferating cells within the TZ of PT cultures seemed to undergo a recovery process with the appearance of chondrocytic clones and extensive ECM–PG production and secretion, which surrounded these cell clusters. Furthermore, the increased mineralization of the inter-territorial matrix between these cells and within the TZ was observed. This might indicate an attempt to restore functional elongation after the loss of normal proliferative chondrocyte due to PT. Hartley et al. [27] and Mizumoto et al. [28] both identified that age as well as the radiation dose had a profound impact on the severity of bone growth implication. In order for bone elongation to take place, chondrocytes must undergo constant mitotic division to replenish the diminishing pool of hypertrophic chondrocytes. This high mitotic activity leaves the growth plate especially vulnerable [10,25]. To analyze the radiosensitivity of GP chondrocytes, Marguliese et al. [29] exposed proliferating chondrocytes to serial radiation doses in vitro. They found a dose-dependent negative effect, e.g., a decrease in proliferation, with increases in cytotoxicity, apoptosis and a disturbance of cell synthetic activity. Horton et al. [10] demonstrated a similar observation after a single 17.5 Gy X-radiation dose. They concluded that this increased proliferative expanse with accelerated ECM synthesis is the primary mode through which bone elongation is re-established after IR. We also observed ECM-specific changes due to PT. The hypertrophic chondrocyte marker ColX was significantly reduced due to PT within the IHC and the gene expression analysis. On the contrary, Col2α1 increased within the IHC and the gene expression studies underlying the altering effect of PT on postnatal bone growth.

While the BMP family are major regulators during cartilage development and bone growth, BMP-2 is particularly important for ongoing developmental processes [5]. We noted an initial decrease in positive BMP-2 and Runx2 staining due to PT after 1 DIV. At 14 DIV, only a profound increase for BMP-2 staining was observed, while PT had a limiting effect. For Runx2, at 14 DIV, only an increase within the non-IR bone slices was noted. IHC data further correlated with the gene expression analysis for BMP-2 and Runx2. Keller et al. [30] found that the reduced BMP-2 signaling within the postnatal GP led to decreased chondrocytes’ proliferation and the shortening of pre-hypertrophic and hypertrophic zones. Studies on BMP expression in postnatal GP (7-day-old rats) conducted by Garrison et al. [31] and Nilsson et al. [32] demonstrated an expression gradient for BMP-2 within the GP. They suggest that, while BMP-2 will stimulate chondrocyte proliferation within the MZ, it will initiate hypertrophic differentiation in the pre-hypertrophic zone. BMP-2 signaling can downstream induce the transcription factor Sox9 or Runx2. Whereas Sox9 will lead MSCs to differentiate towards proliferating chondrocytes, Runx2 will initiate pre-hypertrophic chondrocytes to mature into terminal hypertrophic chondrocytes [33,34]. Our data suggest that growth defects after PT may be due to the decreased BMP-2 and thus downstream Runx2 signaling. This suggests that there is a loss of MSCs which differentiate into proliferating chondrocytes, as well as a loss of pre-hypertrophic chondrocytes, which are responsible for bone elongation. This leads to the disruption or loss of bone elongation, resulting in growth abnormalities.

Bone homeostasis is achieved by osteoblast, which deposit osteoid and ensure bone formation, and osteoclasts, which resorb bone [35]. An imbalance between osteoclast and osteoblast numbers and activity after RT is thought to be one reason for the observed weaker bone structures with increased rates of necrosis or fractures [36]. Therefore, we studied the effects of PT on osteoclast and osteoblast activation and function. Osteoclast activity is regulated through the OPG/RankL signaling pathway. While OPG prevents osteoclast activation, the binding of RankL stimulates their activation [37]. We found that although OPG staining was not as widely distributed within the epiphysis, the staining intensity profoundly increased at 14 DIV. This finding was confirmed at the gene expression level, especially after C-ion IR, in turn causing a highly significant reduction in the OPG expression. Studies found that the OPG produced by hypertrophic chondrocytes is central for the differentiation of osteoclasts/chondroclasts [38,39,40]. In our OTCs, PT had no effect on the development of osteoblasts. However, we found that this was due to the PT gene expression of ALPL being significantly downregulated after 14 DIV. This is in line with the results of other studies, demonstrating that even though osteoblast numbers remain stable most of the time, the functionality of osteoblasts, e.g., osteoid production and mineralization process, is affected due to IR [36,41,42,43].

## 5. Conclusions

In conclusion, our organotypic ex vivo culture of femoral slices provides an excellent method to study pathology and the corresponding regeneration processes after PT in vivo. We observed a significant decline in proliferating chondrocytes at 1 DIV, followed by attempts at regeneration through the formation of chondrocytic clusters at 14 DIV. Furthermore, we noted accelerated mineralization, which correlated with the increased production and secretion of proteoglycans into the pericellular matrix. The expression of Col2α1, which increased during the cultivation period, was significantly inhibited by PT. Moreover, compared to the non-IR control, the decrease in ColX expression over time was more pronounced. Notably, the chondrogenic markers BMP2, RUNX2, OPG, and the osteogenic marker ALPL showed a significant reduction in the increase in expression after 14 DIV as a result of PT. It is important to highlight that C-ions had a more pronounced impact than protons. These novel findings will improve the understanding of cellular radiobiological processes of PT, which will, in turn, contribute to an optimized therapy for tumor patients.

## Figures and Tables

**Figure 1 cells-12-02301-f001:**
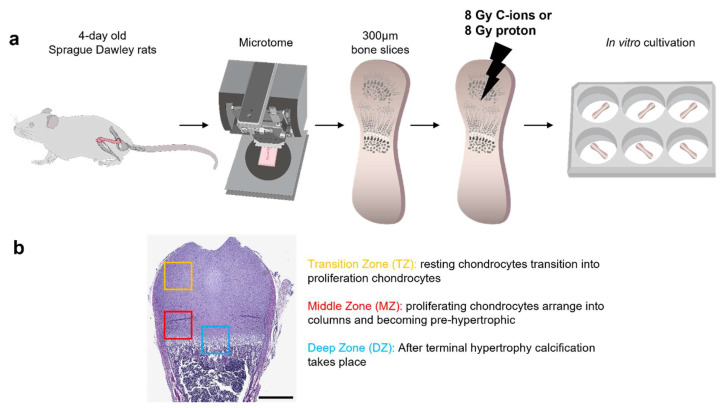
Workflow and respective zones for histological and immunohistochemistry analysis. (**a**) Femurs of 4-day-old Sprague Dawley rats were used for the preparation of the 300 µm ex vivo bone cultures. Bone cultures underwent either 8 Gy proton or C-ions IR followed by 1 or 14 DIV cultivation. (**b**) To assess the influence of the radiation on endochondral ossification, epiphysis was divided based on the underlying stage of chondrocyte maturation and differentiation into the transition zone (TZ), middle zone (MZ), and deep zone (DZ).

**Figure 2 cells-12-02301-f002:**
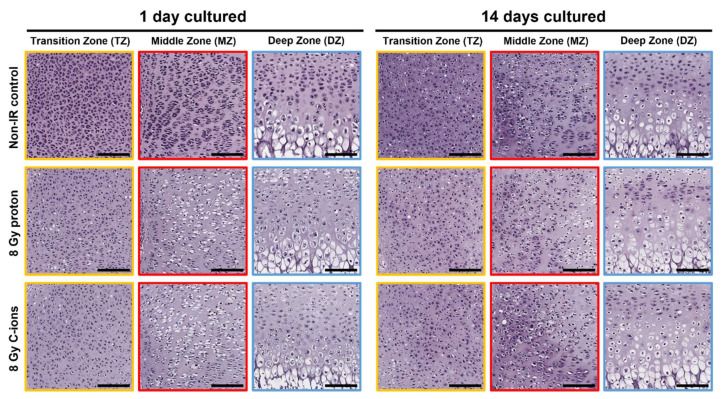
Radiation alters endochondral ossification processes. Respective images of hematoxylin and eosin (H&E) staining from ex vivo bone cultures exposed to either 8 Gy proton or C-ions IR with their respective controls followed by subsequent 1 and 14 DIV cultivation. Colored squares represent the transition zone (TZ, yellow), middle zone (MZ, red), and deep zone (DZ, blue).Total animals used for the histological and IHC analysis were 12 and 3 sections per OTC at different dorso-ventral depths were stained at the same time. Scale bar = 150 µm.

**Figure 3 cells-12-02301-f003:**
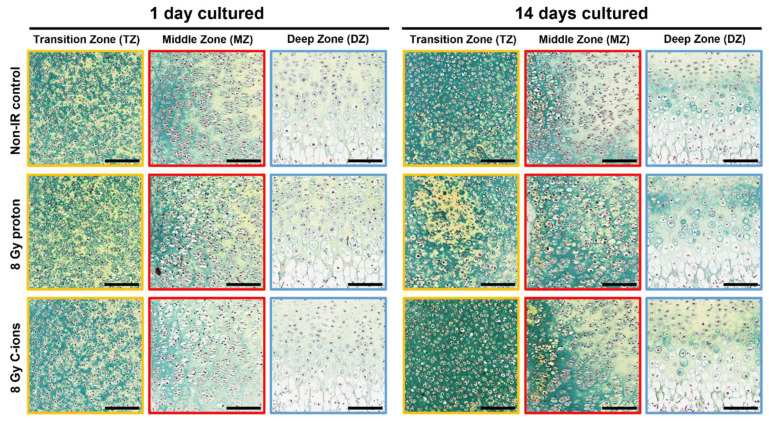
Effect of radiation on tissue development and mineralization. Representative images of Movat’s pentachrome staining of ex vivo bone cultures exposed to either 8 Gy proton or C-ions IR with their respective controls followed by subsequent 1 and 14 DIV cultivation. Colored squares represent transition zone (TZ, yellow), middle zone (MZ, red) and deep zone (DZ, blue). Cellular structures were stained as follows: cell nuclei (black), cytoplasm (reddish), elastic fibers (red), collagen connective tissue (light yellow), mineralized tissue/bone (bright yellow), mineralized cartilage (green). Total animals used for the histological and IHC analysis were 12 and 3 sections per OTC at different dorso-ventral depths were stained at the same time. Scale bar = 150 µm.

**Figure 4 cells-12-02301-f004:**
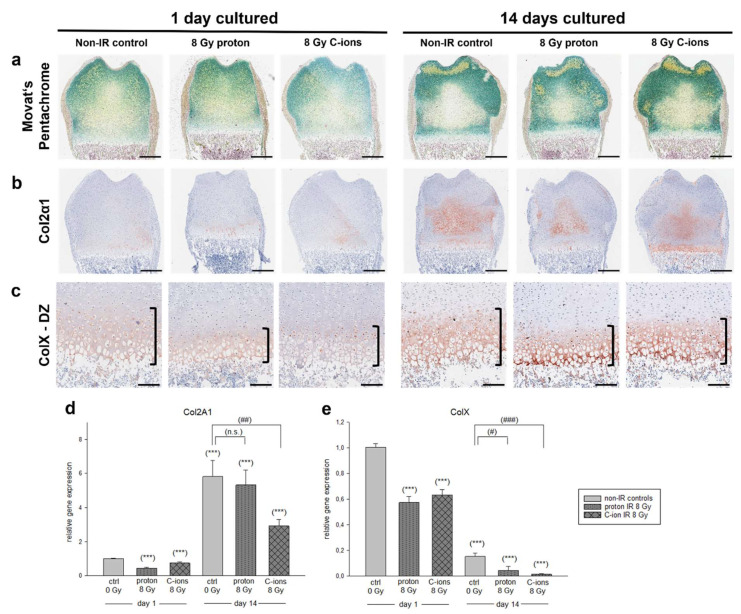
Effect of particle irradiation on extracellular matrix (ECM) composition. (**a**) Representative epiphysis of Movat’s pentachrome staining exposed to either 8 Gy proton or C-ions IR with their respective controls followed by subsequent 1 and 14 DIV cultivation. Cellular structures were stained within Movat’s pentachrome as follows: cell nuclei (black), cytoplasm (reddish), elastic fibers (red), collagen connective tissue (light yellow), mineralized tissue/bone (bright yellow), mineralized cartilage (green). Scale bar = 500 µm. (**b**,**c**) Chondrogenesis is marked by Col2α1 (committed chondrocytes) and ColX (hypertrophic chondrocytes) and was analyzed with IHC staining. Total animals used for the histological and IHC analysis were 12 and 3 sections per OTC at different dorso-ventral depths were stained at the same time. Scale bar = 500 µm. (**d**,**e**) gene expression of Col2α1 and ColX was analyzed by RT-PCR and the expression ratio was calculated with the 2^−ΔΔCt^ method. Results are presented as the mean  ±  SEM; *p* ≤ 0.001 (***) represent the significant differences among day 1 non-IR controls (ctrl); *p*  ≤  0.05 (#), *p*  ≤  0.01 (##), and *p* ≤ 0.001 (###) represent the significances among the 14-day samples. (n.s.) means non-significant differences. In total, 16 animals were used in the gene expression analysis.

**Figure 5 cells-12-02301-f005:**
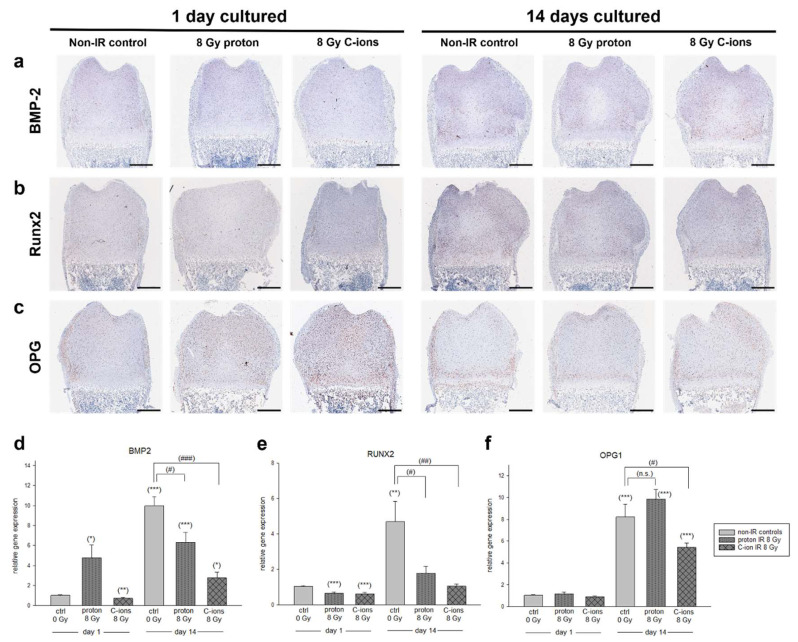
Cartilage regeneration and radiation influence chondrogenesis. (**a**–**c**) Representative images of IHC staining of ex vivo bone cultures exposed to either 8 Gy proton or C-ions IR with their respective control. BMP-2, Runx2, and OPG marking different regeneration and maturation processes. Total animals used for the histological and IHC analysis were 12 and 3 sections per OTC at different dorso-ventral depths were stained at the same time. Scale bar = 500 µm. (**d**–**f**) Gene expressions of BMP-2, Runx2, and OPG were analyzed by RT-PCR and the expression ratio was calculated with the 2^−ΔΔCt^ method. Results are presented as the mean  ±  SEM; *p*  ≤  0.05 (*), *p*  ≤  0.01 (**), and *p* ≤ 0.001 (***) represent the significant differences among the day 1 non-IR controls (ctrl); *p*  ≤  0.05 (#), *p*  ≤  0.01 (##), and *p* ≤ 0.001 (###) represent the significances among the 14 days samples. (n.s.) means non-significant differences. In total, 16 animal were used in the gene expression analysis.

**Figure 6 cells-12-02301-f006:**
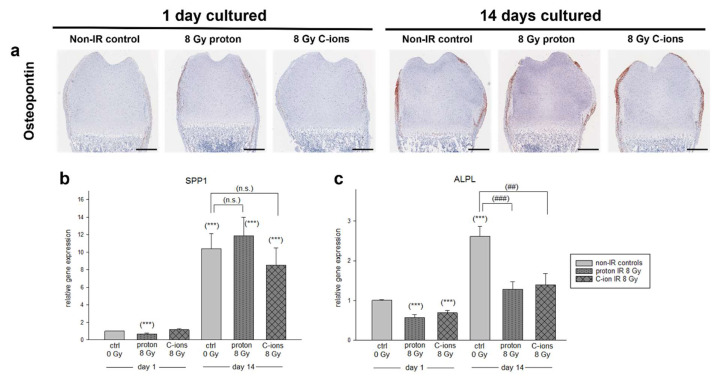
Influence of radiation on osteogenesis. (**a**) Representative images of osteopontin (SPP1) IHC staining of ex vivo bone cultures exposed to either 8 Gy proton or C-ions IR with their respective control. Total animals used for the histological and IHC analysis were 12 and 3 sections per OTC at different dorso-ventral depths were stained at the same time. Scale bar = 500 µm. Gene expression of (**b**) SPP1 and (**c**) ALPL was analyzed by RT-PCR and the expression ratio was calculated with the 2^−ΔΔCt^ method. Results are presented as the mean  ±  SEM; *p* ≤ 0.001 (***) represent the significant differences among day 1 non-IR controls (ctrl); *p*  ≤  0.01 (##), and *p* ≤ 0.001 (###) represent the significances among the 14-day samples. (n.s.) means non-significant differences. In total, 16 animals were used in the gene expression analysis.

## Data Availability

All available data are presented in this article and the corresponding supplementary data.

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
