# Peer review of "Proton and Carbon Ion Irradiation Changes the Process of Endochondral Ossification in an Ex Vivo Femur Organotypic Culture Model"

_cells, 2023, doi:10.3390/cells12182301_

Round 1

Reviewer 1 Report

The manuscript “Proton and carbon ion irradiation changes the process of endochondral ossification in an ex-vivo femur organotypic culture model” touch a very interesting aspect of the effects of particle therapy on normal tissues. The Authors focused their attention on histology, immunohistochemistry, and gene expression  analysis to evaluate these effects on an ex vivo proximal femur 3D organotypic cultures.

I have only one concern on this work. The Authors have very well reported the effects of protons and carbon ion irradiation on their 3D culture model, however it would be really very interesting to have also data with X-rays irradiation on the same experimental model as comparison. In the discussion they refer to some photon data from literature, however these data come from different experimental models and very different doses.

Then, I would suggest to explain the reason for choosing 8 Gy as test dose for both protons and carbon ion irradiation.

An advice from my side: the addition of a table summarizing all the endpoints analysed and the results (stated in a straightforward manner like “+” or “++” or “-“ etc) in all the conditions examined (1 DIV, 14 DIV, 0 Gy, protons, carbon ions) would help the reading and would underlie the importance of the results.

Finally, I have  very few minor concerns:

-        In the abstract, line 18: the sentence starting with “One” is missing of some text;

-        Pag.2, line 74: I assume the Authors were meaning “radioresistant” instead of “radiosensitive”;

-        Pag 3, line 118: “15 DIV” should be a typos for “14 DIV” ;

-        Pag 11, line 375: please add references  for the “well described“

       Pag 13, line 467: the acknowledgments statement is not complete.

In conclusion, I think that this work meets the high standards of the Journal and I reccomend its publication with the revisions I suggested. 

Author Response

The manuscript “Proton and carbon ion irradiation changes the process of endochondral ossification in an ex-vivo femur organotypic culture model” touch a very interesting aspect of the effects of particle therapy on normal tissues. The Authors focused their attention on histology, immunohistochemistry, and gene expression  analysis to evaluate these effects on an ex vivo proximal femur 3D organotypic cultures.

I have only one concern on this work. The Authors have very well reported the effects of protons and carbon ion irradiation on their 3D culture model, however it would be really very interesting to have also data with X-rays irradiation on the same experimental model as comparison. In the discussion they refer to some photon data from literature, however these data come from different experimental models and very different doses.

Authors response: We agree with the reviewer that the irradiation-studies available in the literature have a variety of different study designs. Especially in the field of radiological research, the spectrum is particularly wide. In our OTC follow-up study, the comparison with conventional X-ray irradiation is an already established component.

Concerning the present study: Since a large number of pediatric and adolescent tumor patients receive particle therapy at MedAustron, the focus of this study was to investigate the effects of proton and C-ion irradiation.

Then, I would suggest to explain the reason for choosing 8 Gy as test dose for both protons and carbon ion irradiation.

Authors response: In clinical practice, the particles are applied in the following dosages: 70 Gy in 35 fractions at 2 Gy RBE/fraction (proton IR) and 70.4 Gy in 16 fractions at 4.4 Gy RBE/fraction (C-ions).

However, for cellular research purposes, fractional application is not feasible. The irradiation of cells or bones is typically performed with doses of 2 Gy, 4 Gy, or 8 Gy for both proton and C-ion irradiation. Due to the limited availability of irradiation time for research, which is usually limited to weekends when the particle accelerator is not required for patients, the 8 Gy dosage has been chosen in this particular case to clearly demonstrate the effects being investigated.

An advice from my side: the addition of a table summarizing all the endpoints analysed and the results (stated in a straightforward manner like “+” or “++” or “-“ etc) in all the conditions examined (1 DIV, 14 DIV, 0 Gy, protons, carbon ions) would help the reading and would underlie the importance of the results.

Authors response: Since it is difficult to quantify the results of histological and immunohistological staining in particular, I would like to dispense with the table. Instead of this, we have formulated the conclusion more precisely. It now reads as follows: " In conclusion, our organotypic ex-vivo culture of femoral slices provides an excellent method to study pathology and corresponding regeneration processes after PT in-vivo. We observed a significant decline in proliferating chondrocytes at 1 DIV, followed by attempts at regeneration through the formation of chondrocytic clusters at 14 DIV. Furthermore, we noted accelerated mineralization, which correlated with increased production and secretion of proteoglycans into the pericellular matrix. The expression of Col2α1, which increased during the cultivation period, was significantly inhibited by PT. Moreover, compared to the non-IR control, the decrease in ColX expression over time was more pronounced. Notably, the chondrogenic markers BMP2, RUNX2, OPG, and the osteogenic marker ALPL showed a significant reduction in the increase of expression after 14 DIV as a result of PT. It is important to highlight that C-ions had a more pronounced impact than protons. These novel findings will improve the understanding of cellular radiobiological processes of PT which will in turn contribute to optimize therapy for tumor patients.”

Finally, I have  very few minor concerns: 

In the abstract, line 18: the sentence starting with “One” is missing of some text;

Authors response: the rephrased sentence is as follows: “After irradiation, histological staining, immunohistochemical studies, and gene expression studies were conducted following one or 14 days of in vitro culture (DIV).“

Pag.2, line 74: I assume the Authors were meaning “radioresistant” instead of “radiosensitive”;

Authors response: Oh god, how embarrassing. You are right, of course. The error has been corrected

Pag 3, line 118: “15 DIV” should be a typos for “14 DIV” ;

Authors response: Thank you for the hint. The error has been corrected.

 Pag 11, line 375: please add references  for the “well described“

Authors response: An additional reference to the already listed References #10, 11 and 26 has been added.

Pag 13, line 467: the acknowledgments statement is not complete.

Authors response: oops, the back part of the sentence got lost. We have corrected it.

In conclusion, I think that this work meets the high standards of the Journal and I reccomend its publication with the revisions I suggested. 

Authors response: We would like to thank the reviewer very much for his/her efforts and time and hope that we have addressed all points to his/her satisfaction.

Reviewer 2 Report

The authors present a new method to assess the effect of particle therapy on pediatric bone. The study is very well designed and I could only find some minor points, which should be adressed:

Why did you use the same dose of 8 Gy for both proton and carbon irradiation? Please add some comment in the manuscript.

In the methods sections you writh 15DIV but actually it should be 14 DIV correct?

The sentences in line 229 ff are confusing me. You are writing that C-ion irradiation predominantly causes cartilage to be the dominant tissue. But in the next sentence you write that it is for proton and not C-ion. Could you please clarify.

In figure 4 you are using #, ## and ### for assessing significance I guess. Please provide a description for that in the caption.

The quality of the english writing is quite good. Some better formulations could be found.
Particularly in the abstract (line 39) it should be relation instead of relatation.

Author Response

The authors present a new method to assess the effect of particle therapy on pediatric bone. The study is very well designed and I could only find some minor points, which should be adressed:

Authors response: We would like to thank you very much for the positive assessment of our work. We will make an effort to improve our work even further with your recommendations.

Why did you use the same dose of 8 Gy for both proton and carbon irradiation? Please add some comment in the manuscript.

Authors response: We have added the following sentence at the beginning of the Results section: “To ensure a direct comparison between protons and C-ions, a dose of 8 Gy was selected for both particle types.”

In the methods sections you writh 15DIV but actually it should be 14 DIV correct?

Authors response: Thank you for the hint. The error has been corrected.

The sentences in line 229 ff are confusing me. You are writing that C-ion irradiation predominantly causes cartilage to be the dominant tissue. But in the next sentence you write that it is for proton and not C-ion. Could you please clarify.

Authors response: The first two sentences refer to the TZ (transition zone): “After 14 DIV, bone cultures exposed to proton IR exhibited mineralized tissue (bright yellow) within the TZ compared to the respective control. Contrary, cartilage is the predominant tissue within C-ions IR bone cultures (green).“ In the following sentence the reference to the MZ (middle zone) was missing. We have corrected this error and thank you for pointing it out. The corrected sentence is as follows: „In the area of MZ, cartilage (green) is predominantly presented after proton IR, whereas C-ions IR cultures have a similar distribution as seen in the control.“

In figure 4 you are using #, ## and ### for assessing significance I guess. Please provide a description for that in the caption.

Authors response: The missing explanations regarding the significances were added in the legends of Figures 4, 5, and 6 (this concerns all gene expression analyses).

The quality of the english writing is quite good. Some better formulations could be found.
Particularly in the abstract (line 39) it should be relation instead of relatation.

Authors response: The typo has been corrected.

We would like to thank the reviewer very much for his/her efforts and time and hope that we have addressed all points to his/her satisfaction.

Reviewer 3 Report

First of all, I would like to congratulate the authors to this really nice work!

However, I do have some concerns regarding the paper:

Is there a specific reason why the authors chose 8Gy as their irradiation dose? An explanation as to why this specific dose was chosen should be implemented in the text.

There is no ethics approval number given, while there are no in vivo experiments a “Tierversuchsanzeige” with corresponding approval number should be included.

I am also missing a “Statistical analysis” part in the Methods section. Also, in the Figures no indication as to how statistical significance was calculated is given. This will need to be added for sure.

Another point that is lacking is. Are there no quantification data for all the extensive staining images the authors have carried out? By only seeing a fraction of all stainings (as representative images) and no quantification I find it very hard to judge the soundness of the scientific output. I would thus suggest to carry out a quantification of these data.

While I do think the manuscript is very nice, I do not think that qPCR Data alone is sufficient to undermine the conclusions.

With regards to language: English language is very nice, and easy to read, just a few minor spell checks necessary in my opinion e.g. line 58 page 2 I believe instead of “other” it should read “order”?

Round 2

Reviewer 3 Report

First of all, I would like to thank the authors for taking my comments into consideration. I am, however, still unsure about 2 factors: 

1) The authors explained that, in patient treatment, particles are usually delivered in "70 Gy in 35 fractions at 2 Gy RBE/fraction (proton IR) and 70.4 Gy in 16 fractions at 4.4 Gy RBE/fraction (C-ions)" The also explained that usually, in experiments 2, 4, or 8 Gy are chosen. In my own experiments I also use a lower dose as a full fractionation scheme is not feasible in either animals or ex vivo cultures, usually the single dose that is applied per fraction or the BED is chosen. However, the dose chosen by the authors resembles 2 fractions for C-Ions and 4 fractions for proton IR.  Further, they (understandably!) explain that "Due to the limited availability of irradiation time for research, which is usually limited to weekends when the particle accelerator is not required for patients, the 8 Gy dosage has been chosen in this particular case to clearly demonstrate the effects being investigated" 

However, in the manuscript none of that is mentioned but instead it is stated that: "The 8 Gy dosage has been selected to effectively showcase its impact on postnatal bone elongation within the proximal femur epiphysis" Which is not the same explanation as given in the point-to-point reply. 

I am aware that it is impossible to carry out an exact irradiation scheme as it is done in patients. But there must be a reason (previous data for example?) that particularly 8 Gy was chosen and not e.g. 4 Gy. This is what I was referring to in my previous report.

2) Concerning the quantification of histological and immunohistochemical staining: I am aware that there will always be size differences between pubs and in general in animal experiments. However, the field of view should always be the same size regardless, and per chance, this way the size differences should even out. Also, this is why you use more than one animal so the size and staining differences will be leveled out when you do quantification.

It is, of course, possible to use exemplary images, however, if you chose to only show images without any kind of quantification you cannot draw any conclusions from these images. If you take weight and size differences into consideration, you must also take into consideration that a staining will never be 100% reproducible from slide to slide. Thus, depending on what image you chose to show, you can always get a different output. Of course, the impressions the images give are supported by qPCR data. However, you can not carry over qPCR data 1:1 to stained slides. Also, not in IHC, as in the staining you stain for the protein and in qPCR you look for gene expression which does not always have to be the same at the same timepoint.

Thus, if you do not quantify these images, you either need to show all slides or, in my opinion, you must remove all conclusions you draw from the subjective observation of these slides.

For example: lines 297-300: ”After 14 DIV a profound increase in ColX staining intensity was observed within all conditions. However, PT led to a lower extent of the hypertrophic area. Notably,  at the site of calcification the proton radiated bone slices demonstrated an intensely positive ColX staining.”

Or

Lines 329 -332: After 14 DIV chondrocytes were more intensely stained for BMP-2 for under all conditions. Within the TZ, a reduction of in positive staining was observed due to C-ions IR. A decrease of positive chondrocyte staining was observed within the MZ and DZ for both PT conditions respectivley.” (there is also a minor spelling mistake in this sentence: “respectivley” instead of “respectively”, only saw this because I used it as an example)  

You can also not use this subjective data in the discussion without proper quantification as done here for example:

Lines 439-442 “At 14 DIV only a profound increase for BMP-2 staining was observed, while PT had a limiting effect. For Runx2 at 14 DIV only an increase within the non-IR bone slices was noted. IHC data further correlated with the gene expression analysis for BMP-2 and Runx2.”

3) Also, looking at this now: How many animals in total were used for experiments. You mention that experiments were conducted either at least 3 or 4 times. However, unless I have missed it, there is no number of animals given that have been used per experiment or in total. Please make sure to also add the number of animals per experiment as well as total animals used for both, material and methods section, as well as underneath the respective figures.

Thank you very much! 

Kind regards 

Minior spell check required, as for example, pointed out above.

Author Response

We would also like to thank the reviewer for taking the time to help improve our manuscript.

First of all, I would like to thank the authors for taking my comments into consideration. I am, however, still unsure about 2 factors: 

1) The authors explained that, in patient treatment, particles are usually delivered in "70 Gy in 35 fractions at 2 Gy RBE/fraction (proton IR) and 70.4 Gy in 16 fractions at 4.4 Gy RBE/fraction (C-ions)" The also explained that usually, in experiments 2, 4, or 8 Gy are chosen. In my own experiments I also use a lower dose as a full fractionation scheme is not feasible in either animals or ex vivo cultures, usually the single dose that is applied per fraction or the BED is chosen. However, the dose chosen by the authors resembles 2 fractions for C-Ions and 4 fractions for proton IR.  Further, they (understandably!) explain that "Due to the limited availability of irradiation time for research, which is usually limited to weekends when the particle accelerator is not required for patients, the 8 Gy dosage has been chosen in this particular case to clearly demonstrate the effects being investigated"

However, in the manuscript none of that is mentioned but instead it is stated that: "The 8 Gy dosage has been selected to effectively showcase its impact on postnatal bone elongation within the proximal femur epiphysis" Which is not the same explanation as given in the point-to-point reply. 

I am aware that it is impossible to carry out an exact irradiation scheme as it is done in patients. But there must be a reason (previous data for example?) that particularly 8 Gy was chosen and not e.g. 4 Gy. This is what I was referring to in my previous report.

Authors Response: We acknowledge and appreciate your perspective. The rationale behind selecting an irradiation dose of 8 Gy was influenced by our prior experiences involving proton and C-ion irradiation of human chondrosarcoma cells (Ref. Lohberger et al. Cellular and molecular biological alterations after photon, proton, and carbon ions irradiation in human chondrosarcoma cells linked with high quality physics data. Int. J. Mol. Sci. 2022; Sep 28;23(19):11464. doi: 10.3390/ijms231911464; Lohberger et al. The ATR Inhibitor VE-821 Enhances the Radiosensitivity and Suppresses DNA Repair Mechanisms of Human Chondrosarcoma Cells. Int. J. Mol. Sci. 2023; Jan 24;24(3):2315. doi: 10.3390/ijms24032315). Additionally, our comprehensive literature review revealed that, in clinical settings, when single-fraction radiotherapy (SFRT) is employed usually a dosage of 1 x 8 Gy is applied. (Ref. Saito et al. Single- Versus Multiple-Fraction Radiation Therapy for Painful Bone Metastases: A Systematic Review and Meta-analysis of Nonrandomized Studies. Adv Radiat Oncol. 2019 Oct-Dec; 4(4): 706–715; Peters et al. Adoption of single fraction radiotherapy for uncomplicated bone metastases in a tertiary centre. Clin Transl Radiat Oncol 2021 Jan 15;27:64-69.

Our primary objective in this research endeavor was to evolve an ex vivo model that facilitates the examination of the consequences induced by proton and C-ion radiation on postnatal bone development. Given our observations of dosage-dependent effects within the cell culture, where radiation doses of 2, 4, and 8 Gy were administered, we postulated that, in the context of the 3D ex vivo femur model, a radiation dosage of 8 Gy would be necessary to distinctly manifest the deleterious effects. This elevated dosage selection was grounded in the intention of discerning and subsequently addressing the detrimental impacts of radiation. We hope that our methodology can facilitate the identification of potential preventive measures. We recognize that we have to implement more quantifiable analysis to better demonstrate the effect of possible treatment options towards preventing the growing bone from negative consequences due to IR therapy.

For a more precise definition of the selected irradiation dose we included the following sentences at the beginning of the discussion section (lines 411-417): “In the clinical practice, the particles are applied in the following dosages: 70 Gy in 35 fractions at 2 Gy RBE/fraction (proton IR) and 70.4 Gy in 16 fractions at 4.4 Gy RBE/fraction (C-ions). The irradiation of cells or bones for research purposes is typically performed with doses of 2 Gy, 4 Gy, or 8 Gy for both proton and C-ion irradiation. Due to the limited availability of irradiation time for research, which is usually limited to weekends when the particle accelerator is not required for patients, the 8 Gy dosage has been chosen in this particular case to clearly demonstrate the effects being investigated.”

2) Concerning the quantification of histological and immunohistochemical staining: I am aware that there will always be size differences between pubs and in general in animal experiments. However, the field of view should always be the same size regardless, and per chance, this way the size differences should even out. Also, this is why you use more than one animal so the size and staining differences will be leveled out when you do quantification.

It is, of course, possible to use exemplary images, however, if you chose to only show images without any kind of quantification you cannot draw any conclusions from these images. If you take weight and size differences into consideration, you must also take into consideration that a staining will never be 100% reproducible from slide to slide. Thus, depending on what image you chose to show, you can always get a different output. Of course, the impressions the images give are supported by qPCR data. However, you can not carry over qPCR data 1:1 to stained slides. Also, not in IHC, as in the staining you stain for the protein and in qPCR you look for gene expression which does not always have to be the same at the same timepoint.

Thus, if you do not quantify these images, you either need to show all slides or, in my opinion, you must remove all conclusions you draw from the subjective observation of these slides.

Authors Response: The concerns the reviewer has raised regarding the quantification of histological and immunohistochemical staining in tissue sections are valid and our team has engaged in extensive discussions to devise optimal strategies for quantifying our dataset. However, due to the rapid evolving of chondrocytes within the distinct zones we tried to quantify the IHC throughout the whole proximal femur. Unfortunately, the data did not truly capture the effects seen within the staining’s. Therefore, while we understand the viewpoint of the reviewer, the protocols undertaken to assess/present the data in our histological and immunohistochemistry figures compared by qRT-PCR data follow comparable approaches frequently found in the literature (Su et al., Osteoblast derived-neurotrophin‑3 induces cartilage removal proteases and osteoclast-mediated function at injured growth plate in rats. Bone, 2018, 116, 232–247. Su et al., Neurotrophin-3 Induces BMP-2 and VEGF Activities and Promotes the Bony Repair of Injured Growth Plate Cartilage and Bone in Rats. J Bone Miner Res, 2016, 31: 1258-1274). Hence, we feel that showing representative images are suitable and build upon prior knowledge sufficiently. We aim to further strengthen our OTC methodology and work to establish methodology that is more quantificational. We hope in the future to include single-cell analysis and use Laser dissection technology to isolate RNA from chondrocytes specific to the distinct zones to further advance our knowledge.

3) Also, looking at this now: How many animals in total were used for experiments. You mention that experiments were conducted either at least 3 or 4 times. However, unless I have missed it, there is no number of animals given that have been used per experiment or in total. Please make sure to also add the number of animals per experiment as well as total animals used for both, material and methods section, as well as underneath the respective figures.

Authors Response: We have added the information about the number of rat pups used in the material and methods section in lines 97-100: “To ensure the robustness and reliability of the obtained results, experiments were conducted the experiment 4 times for gene expression analysis (16 rat pups in total) and 3 times for histological and immunohistochemistry staining’s (12 rat pups in total) with 3 sections per OTC at different dorso-ventral depths).” as well as under the respective figures we added: “Total animals used for the histological and IHC analysis were 12 and 3 sections per OTC at different dorso-ventral depths were stained at the same time.” and/or “Total animal used in the gene expression analysis were 16.”

We again thank the reviewer for his effort and time and hope our responses are to the reviewer’s satisfaction.